# Impact of Effluent from the Leachate Treatment Plant of Taman Beringin Solid Waste Transfer Station on the Quality of Jinjang River

**Pui Mun Chin [1]** , **Aine Nazira Naim [1], Fatihah Suja [1,\*] and Muhammad Fadly Ahmad Usul [2]**

[1]  Department of Civil Engineering, Faculty of Engineering and Built Environment,
    Universiti Kebangsaan Malaysia, Bangi Selangor 43600, Malaysia; cpeiwen96@gmail.com (P.M.C.);
    ainenazira@yahoo.com (A.N.N.)
[2]  Department of Solid Waste Management, Ministry of Housing and Local Government, 51 Persiaran Perdana,
    Presint 4, Putrajaya 62100, Malaysia; p95300@siswa.ukm.edu.my
\*  Correspondence: fati@ukm.edu.my; Tel.: +60-1-9304-2621

**Abstract:** Rapid population growth has contributed to increased solid waste generated in Malaysia. Most landfills that have reached the design capacity are now facing closure. Taman Beringin Landfill was officially closed, so the Taman Beringin Solid Waste Transfer Station was built to manage the relocation, consolidation, and transportation of solid waste to Bukit Tagar Sanitary Landfill. Leachates are generated as a consequence of rainwater percolation through waste and biochemical processes in waste cells. Leachate treatment is needed, as leachates cause environmental pollution and harm human health. This study investigates the impact of treated leachate discharge from a Leachate Treatment Plant (LTP) on the Jinjang River water quality. The performance of the LTP in Taman Beringin Solid Waste Transfer Station was also assessed. Leachate samples were taken at the LTP's anoxic tank, aeration tank, secondary clarifier tank, and final discharge point, whereas river water samples were taken upstream and downstream of Jinjang River. The untreated leachate returned the following readings: biochemical oxygen demand (BOD) (697.50 ± 127.94 mg/L), chemical oxygen demand (COD) (2419.75 ± 1155.22 mg/L), total suspended solid (TSS) (2710.00 ± 334.79 mg/L), and ammonia (317.08 ± 35.45 mg/L). The LTP's overall performance was satisfactory, as the final treated leachates were able to meet the standard requirements of the Environmental Quality (Control of Pollution from Solid Waste Transfer Station and Landfill) Regulation 2009. However, the LTP's activated sludge system performance was not satisfactory, and the parameters did not meet the standard limits. The result shows a low functioning biological treatment method that could not efficiently treat the leachate. However, a subsequent step of combining the biological and chemical process (coagulation, flocculation, activated sludge system, and activated carbon adsorption) helped the treated leachate to meet the standard B requirement stipulated by the Department of Environment (DOE), i.e., to flow safely into the river. This study categorized Jinjang River as polluted, with the discharge of the LTP's treated leachates, possibly contributing to the river pollution. However, other factors, such as the upstream sewage treatment plant and the ex-landfill downstream, may have also affected the river water quality. The LTP's activated sludge system performance at the transfer station still requires improvement to reduce the cost of the chemical treatment.

**Keywords:** activated sludge system; Jinjang River; leachate; leachate treatment plant

## 1. Introduction

According to the Solid Waste and Public Cleansing Management Act 2007, solid waste includes any scrap material or other unwanted surplus substance or rejected product arising from any process;

this substance must be disposed of and considered broken; and the authority requires other material to also be disposed of. Malaysia is a developing country, so there are increasing items that must be disposed of, according to the authorities. Since Malaysia is rapidly developing, the concurrent rapid increase in population growth and economic growth will lead to increased solid waste generation. Malaysia has a total urban population of 20,124,970. This population generated about 24,866 tons of municipal solid waste per day in 2012. However, the total population is expected to grow to 33,769,000, so the rate of municipal solid waste generation is also set to increase to 51,655 tons per day by year 2025 [1].

According to the Solid Waste Management Laboratory Report 2012 (National Solid Waste Management Department), 95 percent of waste is managed via landfill. The report also showed that only 165 out of 292 landfills are still operational in Malaysia. Due to a lack of suitable land and the high cost of building new sanitary landfills, solid waste transfer stations are required to collect the municipal solid waste before the waste can be transferred to landfills located far from the city [2].

Solid waste contains biodegradable organic matter that can be broken down into simpler compounds by anaerobic and aerobic microorganisms, a process that leads to leachate formation [3]. Leachate composition and characteristic depend on factors, such as landfill age, climate, organic matter content, type of waste, degree of compaction, and the availability of moisture and oxygen [3–6]. Leachate is usually pH 6 to pH9, making it alkali. Leachate also contains organic matter, such as biochemical oxygen demand (BOD) and chemical oxygen demand (COD), ammonia nitrogen, suspended solids, phosphorus, and inorganic materials, such as copper, lead, and cadmium [7–9]. Table 1 shows the properties of leachate at different stages.

**Table 1.** Properties of leachate at different stage.

| Types of Leachate | Young | Intermediate | Old |
|---|---|---|---|
| Landfill age (years) | <5 | 5–10 | >10 |
| pH | 6.5–7.5 (7) | 7.0–8.0 (7.5) | 7.5–8.5 (8) |
| Chemical Oxygen Demand(COD) (g/L) | 10–30 (15) | 3–10 (5) | <3 (2) |
| Biochemical Oxygen Demand/Chemical Oxygen Demand (BOD/COD) | 0.5–0.7 (0.6) | 0.3–0.5 (0.4) | <0.3 (0.2) |
| Nitrogen Ammonia (mg/L) | 500–1000 (700) | 800–2000 (1000) | 1000–3000 (2000) |
| Chemical Oxygen Demand/Nitrogen Ammonia (COD/N) | 5–10 (6) | 3–4 (3) | <3 (1.5) |

Data from Millot 1986.

The high concentration of biodegradable and refractory organic and inorganic matter in leachate may cause the pollution [10]. Leachate may affect human health as leachate contains heavy metals such as lead, cadmium, aluminum, copper sulfate, nickel, and zinc that exceed Malaysia's Interim National Water Quality Standards (INWQS) [3]. Leachate could also have a long-term impact on the environment and ecosystem, as the seeping process of leachate through soil will contaminate groundwater and surface water [11]. Leachates are also phytotoxic, with toxicity tests showing semi-chronic toxicity in white mustard seeds and leachate toxicity in duckweed at semi-chronic exposure. This result shows that increased leachate concentration also increases growth inhibition [12]. Therefore, leachate must be treated before it is discharged.

Leachate treatment can be divided into biological treatment, physicochemical treatment [13] or a combination of both [14]. Biological treatment is done using sequencing batch bioreactors, membrane bioreactors, aerated lagoons, and up-flow anaerobic sludge blanket reactors [15]. Meanwhile, physicochemical treatment makes use of flocculation-coagulation, adsorption by activated carbon, chemical precipitation, ion exchange, and chemical oxidation [14]. Treatment combination is an efficient and more suitable to treat leachate because it considers the leachate age, season, climatic conditions, regulation criteria, and pollutant concentration [14]. Treatment combinations have obtained high removal percentages of organic matter from leachate. For example, two-stage treatment

using a sequencing batch reactor and coagulation achieved 84.89%, 91.82%, and 85.81% for COD, total suspended solid (TSS), and color removal efficiency, respectively [16]. Combining the coagulation and solar photo-Fenton processes also removed 70–80% of organic matter in leachate, better than current treatments [17].

The Taman Beringin Solid Waste Transfer Station was built to manage solid waste at minimum cost and to collect solid waste at optimal frequency. It receives approximately 1700 tons of municipal solid waste per day from Kuala Lumpur and has a peak capacity of 270 ton/h. The leachate is generated from the solid waste compaction process [2]. Therefore, the National Solid Management Department built a leachate treatment plant at the solid waste transfer station to address the problem of leachate production.

The final treated leachate was discharged downstream of Jinjang River, located approximately 50 m from Taman Beringin Solid Waste Transfer Station. Jinjang River can be categorized as a slightly polluted river based on Water Quality Index (WQI) [18]. This study aims to investigate the impact of discharged leachate from Taman Beringin Solid Waste Transfer Station on Jinjang River's water quality. The performance of the activated sludge system in the leachate treatment plant (LTP) at Taman Beringin Solid Waste Transfer Station was also evaluated based on leachate pollutant removal percentage, as well as operational parameters, design parameters, and kinetic parameters. The physical and chemical properties of raw leachate were also identified.

## 2. Materials and Methods

### 2.1. Study Site

Taman Beringin Solid Waste Transfer Station (TBSWTS) is located on a 12.9 acres site at Taman Beringin, Jinjang Utara, Kuala Lumpur. The solid waste transfer station was built by Kuala Lumpur City Hall (DBKL). TBSWTS commenced operation in April 2002, aiming to address the high cost and scarcity of land and landfill facilities within the Klang Valley. It was built as a part of a modern solution for waste management for solid waste transfer, consolidation, and transportation to the Bukit Tagar Sanitary Landfill, which is located 70 km to 100 km from the transfer station. Figure 1 illustrates the route from TBSWTS to Bukit Tagar Sanitary Landfill.

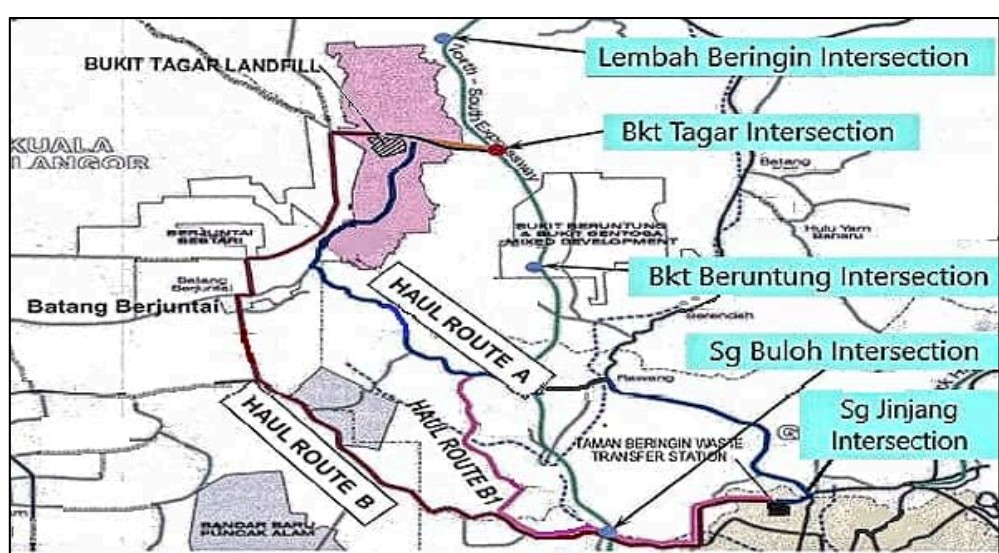

**Figure 1.** Route from Taman Beringin Solid Waste Transfer Station (TBSWTS). Data from National Solid Waste Management Department (2012).

TBSWTS comprises several main facilities, such as a weighbridge waste receiving and waste delivered system, a compaction system, a container semi-trailer, a tractor head, and a prime mover,

and environmental facilities such as a leachate treatment plant, an odor and dust control system, a deodorizer spray system, and a fuel station. Figure 2 shows the horizontal compact transfer station applied in TBSWTS.

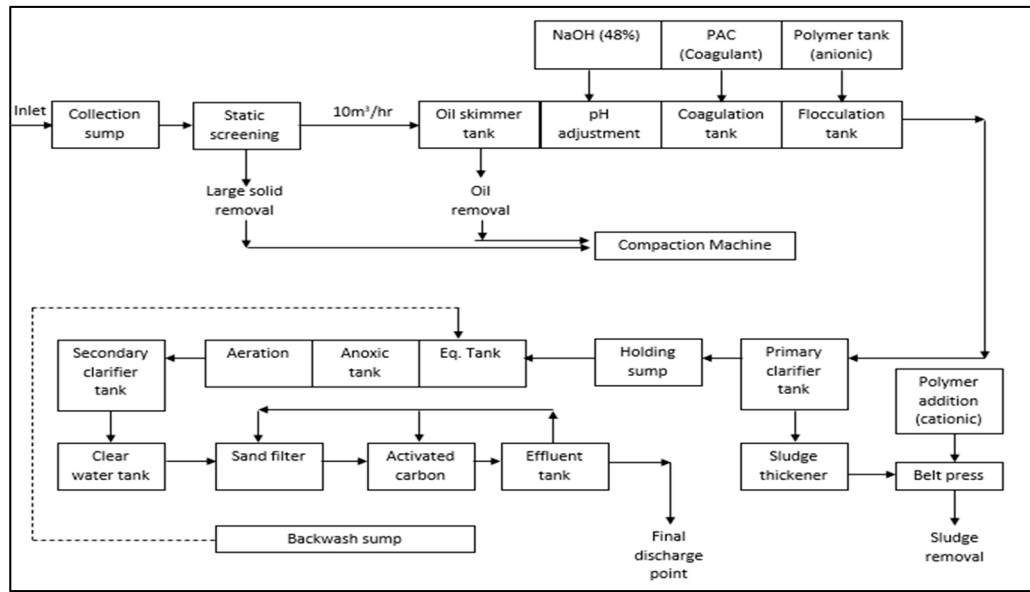

**Figure 2.** System flow in TBSWTS. Data from National Solid Waste Management Department (2012).

The Jinjang River is geographically located at the latitudes of 3°12′20″ N on the equator and 110°40′4″ E on the prime meridian on the Kuala Lumpur map. The river originates from the peak area of Bukit Lagong Recreational Forest in Selangor. It flows in the southeast direction, which houses the districts of Selayang (Selangor) and Jinjang (Kuala Lumpur), and finally ends at a reservoir pond near Taman Beringin before merging with the Klang River. Upstream of the river are villages of indigenous people involved in agriculture and animal husbandry, whereas urbanization and development can mainly be seen downstream of the river [18].

*2.2. Sampling Points*

A site visit was made to TBSWTS and Jinjang River to collect samples and to carry out the analysis. The sampling method was based on the Standard Methods for Examination of Water and Wastewater 23rd Edition [19].

According to National Solid Waste Management Department (JPSPN), the leachate treatment plant operates continuously every day from the collection of leachates up to the discharge of the treated leachate to the river. The treated leachate will be discharged when its volume reaches the maximum indicator level set by JPSPN. The leachate collected from the compactor machine, the oil inceptor, and the washing bay, then flows into the collection sump and undergoes a screening process, pH adjustment, and coagulation and flocculation. The leachate is treated via activated sludge system, a sand filter, and activated carbon adsorption before being discharged into the nearby river. The flow diagram of LTP unit operation is shown in Figure 3.

A total of 500 mL and 1000 mL containers were used to collect the leachate and river water samples, respectively. The samples were collected at around 11 a.m. to 12 noon, four times. For the leachate treatment plant, samples were collected from the anoxic tank (influent), the LTP's aeration tank, secondary clarifier tank (effluent), and final discharge point (final effluent), as shown in Figure 4.

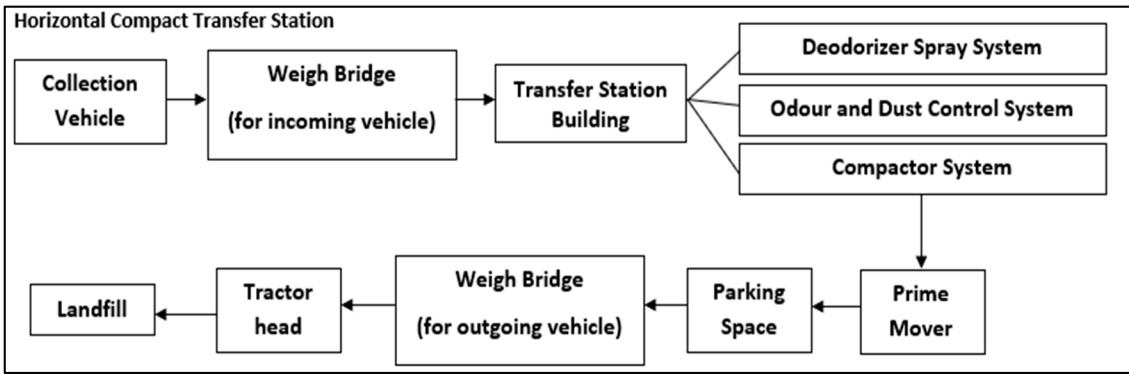

**Figure 3.** Leachate Treatment Plant (LTP) unit operation. Data from National Solid Waste Management Department (2012).

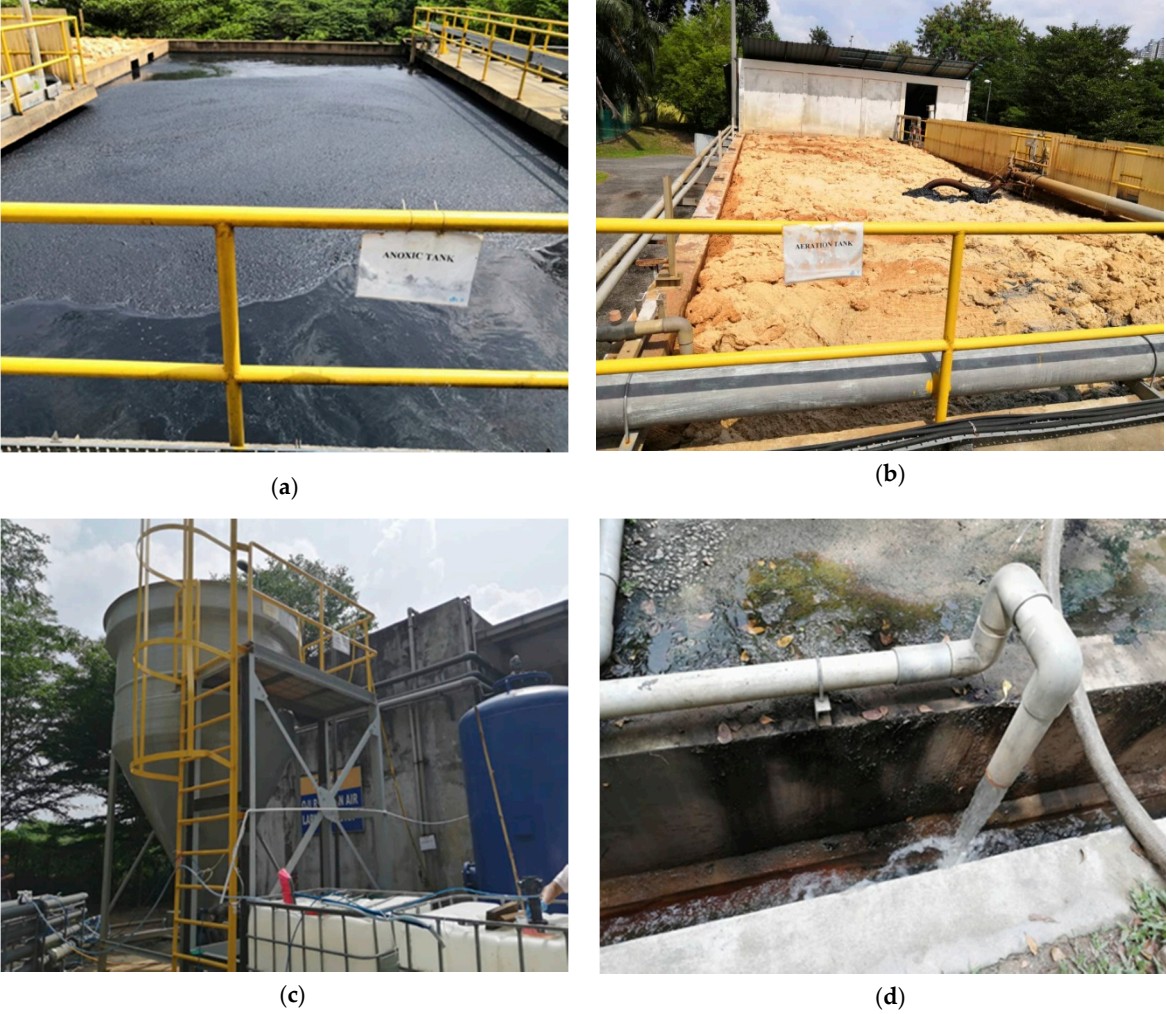

**Figure 4.** Sampling points. (**a**) Anoxic tank, (**b**) aeration tank, (**c**) secondary clarifier tank, (**d**) final discharge point.

The river samples were taken upstream and downstream of Jinjang River approximately 1 km from the Taman Beringin LTP, as shown in Figure 5. The bottles were sealed with parafilm immediately upon the completion of sample collection. To minimize the potential of volatilization and biodegradation between sampling and analysis, the samples were stored at 6 °C, if immediate analysis could not be done.

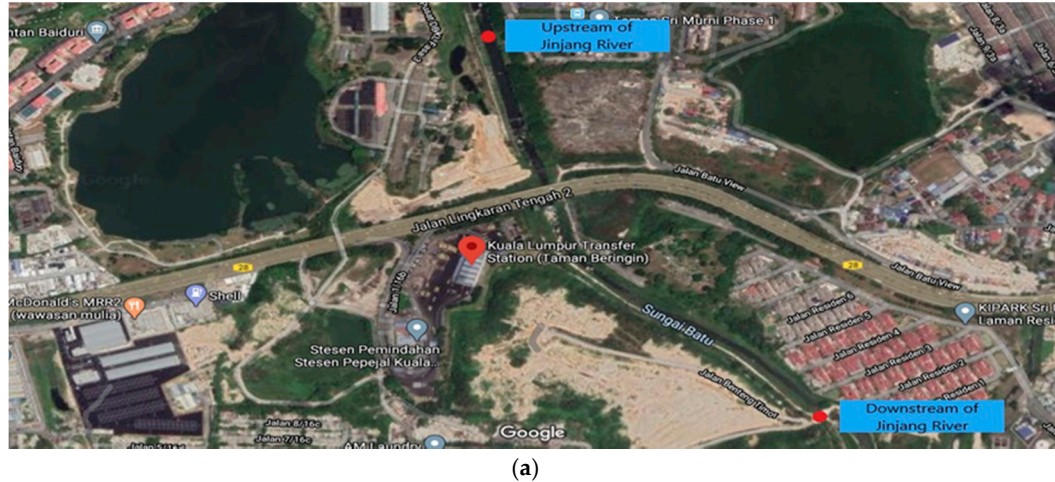

(a)

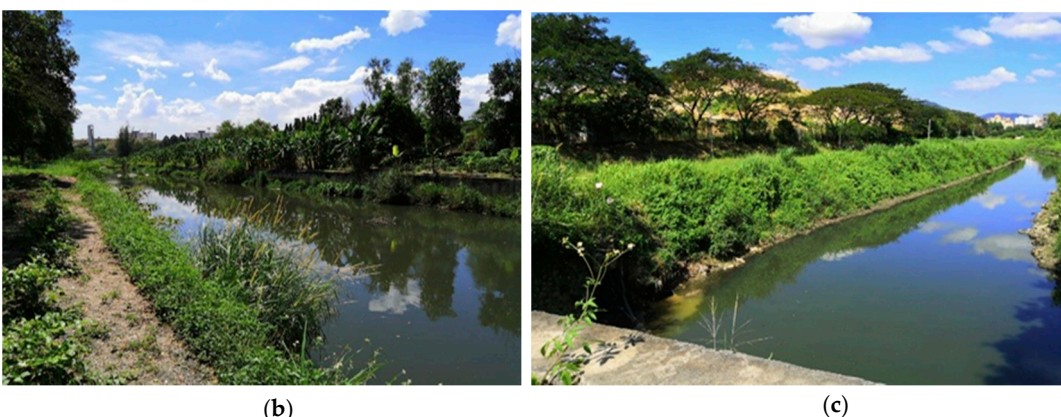

(b)             (c)

**Figure 5.** (**a**) Sampling points at Jinjang River, (**b**) upstream of Jinjang River, (**c**) downstream of Jinjang River.

### 2.3. Experimental Procedures

The samples were analyzed in-situ and via laboratory tests, as summarized in Table 2. The laboratory tests were carried out in accordance with the Standard Methods for Examination of Water and Wastewater 23rd Edition [19] or the Hach method.

**Table 2.** Summary of experimental procedures.

| In-Situ Test | Laboratory Tests |
| --- | --- |
| pH | Biochemical Oxygen Demand ( BOD), (APHA 5210B) |
| Dissolved Oxygen (DO) | Chemical Oxygen Demand (COD) (USEPA Reactor Digestion Method—Method 8000) |
| | Total Suspended Solid, TSS (APHA 2540D) |
| | Ammonia Nitrogen (Salicylate Method—Method 10031) |
| | Total Kjeldahl Nitrogen, TKN (APHA 4500-$N_{org}$B) |
| | Mixed Liquor Suspended Solid, MLSS (APHA 2540E) |

### 2.4. Data Analysis

#### 2.4.1. Parameter of Activated Sludge System

Operational parameters such as pH, dissolved oxygen (DO), mixed liquor suspended solid (MLSS), and COD:N ratio were obtained from in-situ reading and laboratory test results. Those parameters were compared with the activated sludge system standard range to determine the efficiency of LTP's

activated sludge system. Kinetic parameters, such as biomass yield coefficient (Y), endogenous decay coefficient ($k_d$), specific growth rate ($\mu$), and half velocity constant ($K_s$) were obtained from a graph plot of the results of solid retention time (SRT), MLSS, the effluent suspended solids, and the influent and effluent of BOD. From the graph of 1/theta against $(S_0 - S)$/Theta $\times$ X, the Y value is the coefficient of the slope and the $-k_d$ value is the coefficient of the y-intercept. From the graph of theta $\times$ X/Y$(S_0 - S)$ against 1/S, the $\mu_{max}$ value is the 1/(coefficient of y-intercept) whereas the $K_s$ value is the slope $\times$ $\mu_{max}$.

The design parameters include the food-to-microorganisms (F/M) ratio, the solid retention time (SRT) and the organic loading rate (OLR), which can be calculated using Equations (1)–(3), respectively.

$$\text{F/M} = (\text{BOD or COD influent (kg/day)})/(\text{MLVSS in aeration tan k (kg/day)}) \tag{1}$$

$$\text{SRT (days)} = \quad (\text{Solids in aeration tan k (kg)})/(\text{Effluent solids (kg/day)} \\ + \text{Solids wasted (kg/day)}) \tag{2}$$

$$\text{OLR (g} - \text{days/L)} \quad = (\text{Influent substrate concentration (mg/L)} \\ \times \text{Flowrate (L/days)})/(\text{Digester Volume (L)}) \tag{3}$$

### 2.4.2. Jinjang River Water Quality Analysis

The Water Quality Index (WQI) ascribes a quality value to an aggregate set of measured parameters. It usually consists of sub-index values assigned to parameters, such as Dissolved Oxygen, DO (SIDO), Biochemical Oxygen Demand, BOD (SIBOD), Chemical Oxygen Demand, COD (SICOD), Suspended Solid, SS (SISS), ammonia nitrogen (SIAN), and pH (SIPH), by comparing its measurement with a parameter-specific rating curve, which is then optionally weighted, and combined into the final index. Calculations are performed not on the parameters themselves, but on their sub-indices. The best -it equations used for estimating the sub-index values are shown in Table 3.

**Table 3.** Best fit equations for sub-index values.

| Parameter | Sub-Index | |
|---|---|---|
| DO (SIDO) (in % saturation) | $\text{SIDO} = 0$ <br> $\text{SIDO} = 100$ <br> $\text{SIDO} = -0.395 + 0.030x^2 - 0.00020x^3$ | for $x \leq 8$ <br> for $x \geq 92$ <br> for $8 < x < 92$ |
| BOD (SIBOD) | $\text{SIBOD} = 100.4 - 4.23x$ <br> $\text{SIBOD} = 108 \times 10^{-0.055x} - 0.1$ | for $x \leq 5$ <br> for $x > 5$ |
| COD (SICOD) | $\text{SICOD} = -1.33x + 99.1$ <br> $\text{SICOD} = 103 \times 10^{-0.0157x} - 0.04x$ | for $x \leq 20$ <br> for $x > 20$ |
| NH$_3$-N (SIAN) | $\text{SIAN} = 100.5 - 105x$ <br> $\text{SIAN} = 94 \times 10^{-0.573x} - 5\lvert x - 2 \rvert$ <br> $\text{SIAN} = 0$ | for $x \leq 0.3$ <br> for $0.3 < x < 4$ <br> for $x \geq 4$ |
| TSS (SISS) | $\text{SISS} = 97.5 \times 10^{-0.00676x} + 0.05x$ <br> $\text{SISS} = 71 \times 10^{-0.0016x} - 0.015x$ <br> $\text{SISS} = 0$ | for $x \leq 100$ <br> for $100 < x < 1000$ <br> for $x \geq 1000$ |
| pH (SIPH) | $\text{SIPH} = 17.2 - 17.2x + 5.02x^2$ <br> $\text{SIPH} = -242 + 95.5x - 6.67x^2$ <br> $\text{SIPH} = -181 + 82.4x - 6.05x^2$ <br> $\text{SIPH} = 536 - 77.0x + 2.76x^2$ | for $x < 5.5$ <br> for $5.5 \leq x < 7$ <br> for $7 \leq x < 8.75$ <br> for $x \geq 8.75$ |

Note: $x$ = concentration in mg/L for all parameters except pH. Source: National Water Quality Standards for Malaysia.

Once the respective sub-indices have been calculated, the WQI can be calculated using Equation (4).

$$\text{WQI} = 0.22\text{SIDO} + 0.19\text{SIBOD} + 0.16\text{SICOD} + 0.15\text{SIAN} + 0.16\text{SISS} + 0.12\text{SIPH} \tag{4}$$

The result obtained is then compared with the Department of Environment (DOE) WQI classification, as shown in Tables 4 and 5. The INWQS defines the classification of rivers based on the water quality, with Class I being the 'best' and Class V the 'worst'.

**Table 4.** Department of Environment (DOE) Water Quality Index classification.

| Parameters | Unit | Classes | | | | |
|---|---|---|---|---|---|---|
| | | I | II | III | IV | V |
| Ammoniacal Nitrogen | mg/L | <0.1 | 0.1–0.3 | 0.3–0.9 | 0.9–2.7 | >2.7 |
| Biochemical Oxygen Demand (BOD$_5$) | mg/L | <1 | 1–3 | 3–6 | 6–12 | >12 |
| Chemical Oxygen Demand (COD) | mg/L | <10 | 10–25 | 25–50 | 50–100 | >100 |
| Dissolved Oxygen | mg/L | >7 | 5–7 | 3–5 | 1–3 | <1 |
| pH | mg/L | >7 | 6–7 | 5–6 | <5 | >5 |
| Total Suspended Solid (TSS) | mg/L | <25 | 25–50 | 50–150 | 150–300 | >300 |
| Water Quality Index (WQI) | mg/L | >92.7 | 76.5–92.7 | 51.9–76.5 | 31.0–51.9 | <31.0 |

Source: National Water Quality Standards for Malaysia.

**Table 5.** DOE water quality classification based on Water Quality Index.

| Parameters | Index Range | | |
|---|---|---|---|
| | Clean | Slightly Polluted | Polluted |
| SIBOD | 91–100 | 80–90 | 0–79 |
| SIAN | 92–100 | 71–91 | 0–70 |
| SISS | 76–100 | 70–75 | 0–69 |
| WQI | 81–100 | 60–80 | 0–59 |

Source: National Water Quality Standards for Malaysia.

## 3. Results and Discussion

### 3.1. Raw Leachate Characteristics

The physical and chemical characteristics of raw leachate taken from the Taman Beringin Solid Waste Transfer Station LTP are listed in Table 6.

**Table 6.** Raw leachate characteristics.

| No | Parameter | Value (Average ± SD) | Environmental Quality (Sewage) Regulation 2009 |
|---|---|---|---|
| 1 | pH | 7.73 ± 0.08 | 6–9 |
| 2 | Temperature, °C | 32.7 ± 0.79 | 40 |
| 3 | Dissolved oxygen (DO), mg/L | 0.19 ± 0.08 | - |
| 4 | Biochemical Oxygen Demand (BOD), mg/L | 697.50 ± 127.94 | 20 |
| 5 | Chemical Oxygen Demand (COD), mg/L | 2419.75 ± 1155.22 | 400 |
| 6 | Total Suspended Solid (TSS), mg/L | 2710.00 ± 334.79 | 50 |
| 7 | Nitrogen (Ammonia), mg/L | 317.08 ± 35.45 | 5 |
| 8 | Total Kjeldahl Nitrogen (TKN), mg/L | 339.50 ± 94.11 | - |

Raw leachate is black and has an unpleasant odor. According to the Table 6, the pH and temperature of the raw leachate are within the standard range of the Environmental Quality (Control of Pollution from Solid Waste Transfer Station and Landfill) Regulation 2009. However, the DO content in raw leachate was low due to the aerobic microorganism metabolic reactions that degrade the biodegradable matter in the leachate during the solid waste compaction process at the transfer station.

The concentration of BOD$_5$ and COD in raw leachate exceeded the standard limit. The high BOD$_5$ concentration indicates that leachate cannot self-purify excessive organic matter [20], while the high COD value may be attributed to the presence of high levels of pollutants and humic acid substrates

that could not be stabilized by microorganisms [21]. The BOD$_5$/COD ratio for the raw leachate was 0.29, which is in the range of the acetogenic and methanogenic phase [22]. The intermediate value of BOD$_5$/COD ratio may be due to the continuous waste decomposition process [23].

The result shows that the leachate sample still contained high amounts of suspended solids although the sample had undergone the coagulation and flocculation process before entering the anoxic tank. The TSS value of the leachate sample exceeded the standard limit, so it may affect the water quality if gone untreated [24].

The raw leachate in this study had high concentrations of ammonia that exceeded the standard limit. The presence of ammonia may be due to the deamination of amino acids during the decomposition of organic compounds [25]. The high concentration of nitrogen ammonia is a major pollutant in leachate that affects the removal of COD [26]. However, this value is considered lower than that of the leachate from landfill [27,28]. The total Kjeldahl nitrogen (TKN) value was 339.50 ± 94.11 mg/L, indicating the sum of nitrogen bound in organic substances, in ammonia and in ammonium in the raw leachate.

## 3.2. Performance in Removal of Pollutants in Leachate

The concentration of BOD, COD, ammonia nitrogen, and TKN in the influent samples, the effluent samples, and the final effluent samples are plotted in Figure 6a–d respectively.

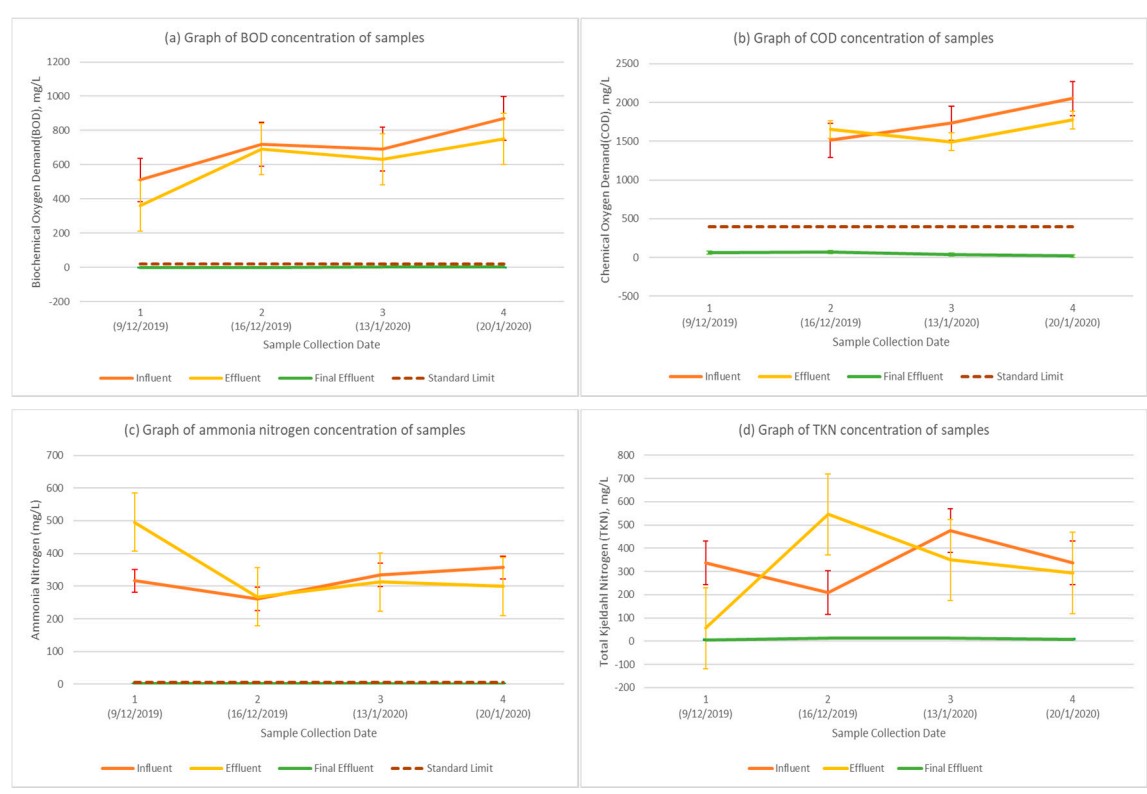

**Figure 6.** Graph of concentration in (**a**) BOD, (**b**) COD, (**c**) ammonia nitrogen, (**d**) TKN.

Figure 6 shows that the activated sludge system did not function efficiently, as the effluent samples still had a high concentration of BOD, COD, ammonia nitrogen, and TKN. The overall pollutant removal percentage in the effluent did not reach 50% or above. However, the final effluent had the least pollutant content, which means that the removal percentage of BOD, COD, ammonia nitrogen and TKN reached 95% and above, indicating that the leachate was completely treated in the activated carbon adsorption process.

According to the Environmental Quality (Control of Pollution from Solid Waste Transfer Station and Landfill) Regulation 2009, the treated leachate must meet standard requirements before it is

discharged to the river. That is, the BOD content must be less than 20 mg/L, the COD content must be less than 400 mg/L, and the ammonia nitrogen content must be less than 5 mg/L. The laboratory results showed that the effluent samples did not meet the above standard requirements, so it can cause pollution. However, the final effluent samples were able to meet the standard requirements and could safely be discharged into the river.

Past research had shown that the adsorption of activated carbon could efficiently remove the organic matter and ammonia, given the appropriate time, agitation speed, temperature, pH, and dosage of activated carbon [29–31]. Due to the inherent physical properties, large surface area, macro structure, high adsorption capacity, and surface reactivity, activated carbon adsorption can efficiently remove pollutants in leachate. The combination of activated sludge and activated carbon could increase pollutant removal efficiency [32]. Therefore, the activated sludge process in the leachate treatment plant of Taman Beringin Solid Waste Transfer Station could not treat leachate, alone, to achieve the standard requirement of the Environmental Quality (Control of Pollution from Solid Waste Transfer Station and Landfill) Regulation 2009.

### 3.3. Health of the Activated Sludge System

#### 3.3.1. Operational Parameter

The laboratory result of leachate samples from the aeration tank are listed in Table 7.

**Table 7.** Operational parameters based on laboratory result.

| No | Parameter | Value (Average ± SD) | Monitoring Point (Fleming 2014a) |
|---|---|---|---|
| 1 | pH | 8.40 ± 0.07 | 6.5–7.5 (Bacteria growth) 7.8–8.2 (Nitrification process) |
| 2 | Dissolved Oxygen (DO), mg/L | 0.16 ± 0.02 | 2–4 |
| 3 | Mixed liquor suspended solid (MLSS), mg/L | 2335 ± 174.24 | 2000–4000 |
| 4 | Chemical Oxygen Demand: Nitrogen (COD:N) | 100:13 | 100:5 |

pH and alkalinity are important parameters that can affect activated sludge microorganisms, especially in the biological treatment process [33]. The result shows the pH measured in-situ to be between pH 8.3 to pH 8.5, which exceeds the optimum pH for bacterial growth or nitrification. The organic matters are decomposed into inorganic matters by microorganisms [34]. Nitrification is a process of converting ammonium ion to nitrate and nitrite ion, as ammonium ion is 20 times more toxic than nitrate ion [35]. However, in this study, bacteria activities were inhibited because the high pH could lead to decreased efficiency of pollutant removal from the effluent. The overall removal percentage of ammonia nitrogen was only 10%, indicating that the nitrification process was affected by the high pH of the activated sludge system.

The dissolved oxygen (DO) concentration in the aeration tank in the activated sludge system plays an important role in treatment efficiency, operating cost and system stability [36]. Each microorganism must have at least (0.1 to 0.3) mg/L DO to metabolize food and reproduce [37]. If the DO content is too low, an unstable environment will result and cause the microorganisms to die, as the anaerobic zones and the sludges have not been properly treated. The result shows that DO concentration in the aeration tank was too low, preventing the activated sludge system from treating the leachate effectively.

MLSS concentration is a measure of the total concentration of solids, including both inert and organic solids, in the aeration tank [33]. The laboratory result showed a standard MLSS concentration. The specific adsorption capacity of organic matter was stable when MLSS increased from (2250 to 2750) mg/L but decreased from (0.17 to 0.105) mgCOD/mgMLSS as the MLSS concentration increased from (2570 to 4500) mg/L [38]. This research supports that particular MLSS concentrations (2000 to 3000) mg/L can maintain the efficiency of the activated sludge system. The volatile suspended

solid (VSS) of the leachate samples from the aeration tank was 1703.03 ± 118.74 mg/L and the VSS/TSS ratio was 0.73, which is within the standard for this research [33].

The result showed that the COD:N ratio was 100:13, but this ratio is higher than rule of thumb stating that COD:N:P should be 100:5:1 to supply enough nutrient (carbon, nitrogen, and phosphorus) for microorganisms. However, past research such as [39,40] proved that a COD:N:P = 100:13:3 could still provide sufficient nutrients for aerobic bacterial growth and a COD:N:P = 100:13:8 could remove 88–92% of COD.

### 3.3.2. Design Parameter

The food-to-microorganism (F/M) ratio of the LTP's activated sludge system was between 0.10 and 0.18 (or average value of 0.13). This ratio did not reach the standard range (0.2 to 0.4) for a conventional activated sludge system [41]. A low F/M ratio indicates that the food supply is limited but there are many microorganisms [33]. Extracellular polymeric substances (EPS) will be produced when bacteria run low on food. This method, which is used to store and concentrated BOD, is similar to that of floc-forming bacteria. This process is beneficial, as flocculation is required to obtain clear effluent. However, the EPS will be degraded once bacteria become low on food for an extended period of time. Then, floc disintegration will begin to result in a cloudy supernatant [42].

The solid retention time (SRT) of the LTP's activated sludge system was between 2 and 4 days. However, a conventional activated sludge system needs about 3–15 days of SRT [43]. The average SRT was 2.69 days, less than 3 days the average SRT, hence resulting in young sludge and poor effluent. In [44], a low SRT (<20 days) caused a membrane bioreactor to fail to effectively remove the carbon and nutrients to treat leachate. A longer SRT may favor the retention and development of microorganisms for better removal refractory organic matter [45]. The above works show that the SRT for a treatment plant's activated sludge system should be extended to increase the leachate treatment efficiency.

The BOD loading rate of a conventional activated sludge system should not exceed 0.04 lbs BOD-days/ft$^3$ (or equivalent to 0.64 kg BOD days/m$^3$) [43]. The result showed a 0.20 kg BOD days/m$^3$ average BOD loading rate for the activated sludge system, which exceeds the standard. The COD loading rate from the result was (0.4–0.6) kg COD-days/m$^3$, but the first sample reached 1.24 kg COD days/m$^3$. The highest COD removal percentage (92.45%) and BOD removal percentage (96%) in SBR were achieved at a loading rate of (0.75–1.5) kg COD days/m$^3$ [46]. The COD removal efficiency could reach 82.3% in a reactor with a 10-day hydraulic retention time (HRT) at 35 °C and a loading rate of 1.0kg COD days/m$^3$ [47]. Therefore, the low organic loading rate of LTP's activated sludge system caused decreased organic matter removal during the leachate treatment process.

### 3.3.3. Kinetic Parameter

Figures 7 and 8 show Taman Beringin Solid Waste Transfer Station LTP's activated sludge system graph of kinetic parameters such as Y, $k_d$, μ, and $K_s$ based on the laboratory results. Table 8 shows the kinetic parameters obtained from the graphs.

**Table 8.** Activated sludge system coefficient of kinetic parameters.

| Coefficient | Unit | Value (This Study) | Value from (Chae et al. 2000) [48] |
|:---:|:---:|:---:|:---:|
| Y | mg VSS/mg BOD$_5$ | 5.655 | 0.36 |
| $k_d$ | day$^{-1}$ | −0.247 | 0.022 |
| $\mu_{max}$ | day$^{-1}$ | 500 | 0.56 |
| $K_s$ | $\frac{mg}{l}$, BOD$_5$ | −0.005 | 612 |

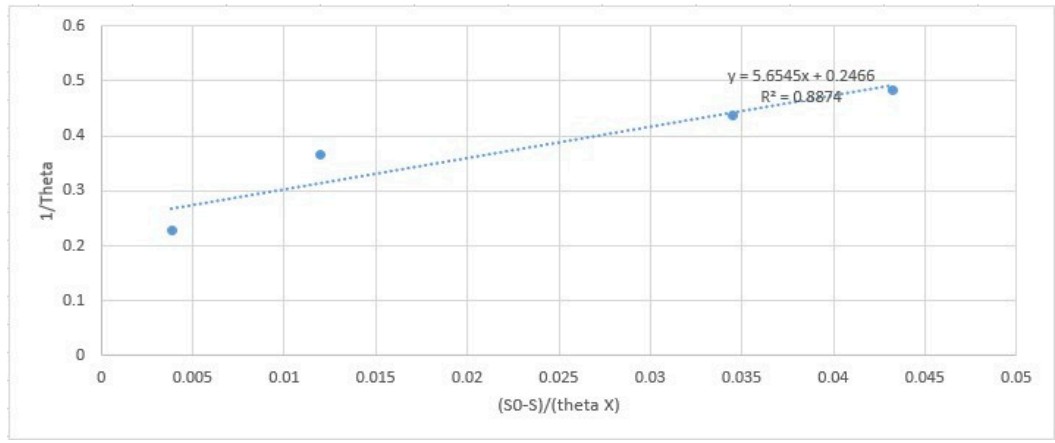

**Figure 7.** Graph of kinetic parameters (Y and $k_d$) of the LTP's activated sludge system.

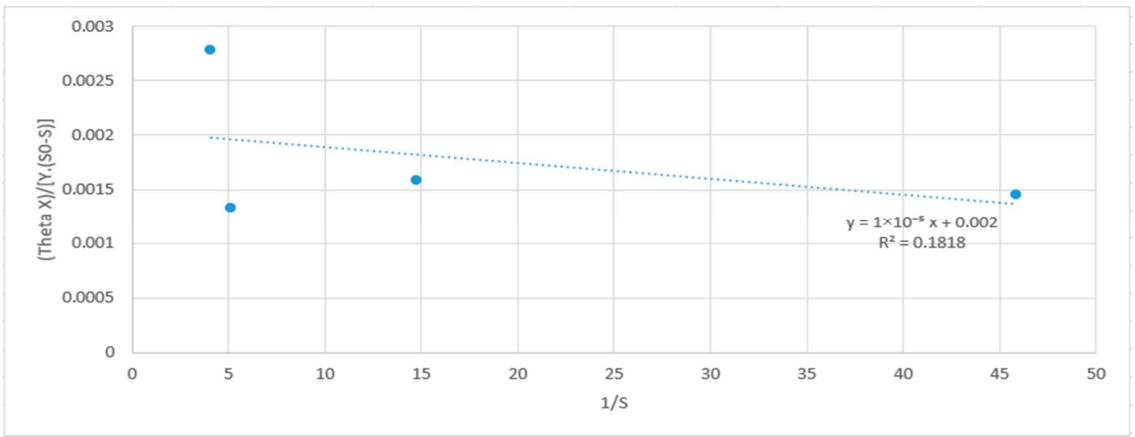

**Figure 8.** Graph of kinetic parameters (($\mu$ and $K_s$) of the LTP's activated sludge system.

The biomass yield coefficient (Y) represents the amount of biomass produced when the substrate is removed by microorganisms [49]. The result shows a higher Y value than that of Chae et al. (2000) [48]. A high Y coefficient indicates that the reactor operates under rich food conditions and that the microorganisms have sufficient food to consume [50]. However, for this study, the high amount of biomass may be due to the remaining organic matter that could not be treated in the activated sludge process. The endogenous decay coefficient ($k_d$) in this study was lower than that of [48]. $k_d$ represents the loss of biomass due to endogenous respiration per unit mass per unit time. A lower $k_d$ value indicates that the microorganisms are active for longer periods in the reactor [49]. It also indicates a lower bacterial decay rate [51].

The maximum specific growth rate ($\mu_{max}$) for this study was much higher than that of benchmark [48]. The low value of $\mu_{max}$ indicates that the substrate is not easily to be biodegraded [49].

The study showed a lower half velocity constant ($K_s$) than the benchmark [48]. The high $K_s$ value indicates that the microorganisms could not decompose the substrate easily [52]. The higher $K_s$ value will lead to a lower biological cell growth rate [49]. Therefore, the lower $K_s$ in this study should indicate the maximum bacterial yields at a low substrate concentration and that the substrates are easily decomposed by the microorganisms. However, the optimal decomposition of substrate was affected by the other parameters, so the overall efficiency being reduced.

### 3.4. Water Quality Analysis of Jinjang River

Tables 9 and 10 show the laboratory results of the river water samples collected upstream and downstream of Jinjang River, respectively.

**Table 9.** Laboratory result (sub-index values) of river water samples upstream.

| Parameters (Sub-Index) | Concentration (mg/L) | | | | | Classes (Index Range) |
|---|---|---|---|---|---|---|
| | Sample 1 (9 December) | Sample 2 (16 December) | Sample 3 (13 January) | Sample 4 (20 January) | Average ± SD | |
| pH | 7.14 | 7.95 | 7.89 | 8.22 | 7.80 ± 0.40 | I |
| (SIPH) | 98.91 | 91.70 | 92.51 | 87.54 | 92.67 ± 4.07 | - |
| Dissolved Oxygen | 2.2 | 2.19 | 2.1 | 2.91 | 2.35 ± 0.33 | IV |
| (SIDO) | 18.57 | 17.66 | 15.33 | 28.26 | 19.96 ± 4.94 | - |
| Biochemical Oxygen Demand, (BOD$_5$) | 0.20 | 0.50 | 0.70 | 0.60 | 0.50 ± 0.19 | I |
| (SIBOD) | 99.55 | 98.29 | 97.45 | 97.86 | 98.29 ± 0.79 | Clean |
| Chemical Oxygen Demand, COD | 23.67 | 46.00 | 23.00 | 19.33 | 28.00 ± 10.52 | III |
| (SICOD) | 67.62 | 37.92 | 68.51 | 75.27 | 62.33 ± 14.40 | - |
| Ammoniacal Nitrogen | 5.27 | 5.70 | 4.80 | 5.00 | 5.19 ± 0.34 | V |
| (SIAN) | 0 | 0 | 0 | 0 | 0 | Polluted |
| Water Quality Index (WQI) | 41.71 | 35.70 | 40.05 | 43.44 | 40.23 ± 2.87 | Polluted |

**Table 10.** Laboratory result (sub-index values) of river water samples downstream.

| Parameters (Sub-Index) | Concentration (mg/L) | | | | | Classes (Index Range) |
|---|---|---|---|---|---|---|
| | Sample 1 (9 December) | Sample 2 (16 December) | Sample 3 (13 January) | Sample 4 (20 January) | Average ± SD | |
| pH | 7.20 | 7.99 | 7.81 | 8.44 | 7.86 ± 0.44 | I |
| (SIPH) | 98.65 | 91.14 | 93.52 | 83.49 | 91.70 ± 5.46 | - |
| Dissolved Oxygen | 4.55 | 3.80 | 3.62 | 5.1 | 4.27 ± 0.59 | III |
| (SIDO) | 58.80 | 43.86 | 39.42 | 67.36 | 52.36 ± 11.25 | - |
| Biochemical Oxygen Demand (BOD$_5$) | 0.60 | 1.90 | 1.00 | 1.50 | 1.25 ± 0.49 | II |
| (SIBOD) | 97.86 | 92.36 | 96.17 | 94.06 | 95.11 ± 2.08 | Clean |
| Chemical Oxygen Demand, COD | 37.00 | 65.67 | 27.33 | 34.00 | 41.00 ± 14.67 | III |
| (SICOD) | 49.89 | 11.76 | 62.75 | 53.88 | 44.57 ± 19.51 | - |
| Ammoniacal Nitrogen | 4.97 | 4.30 | 3.70 | 5.80 | 4.69 ± 0.78 | V |
| (SIAN) | 0 | 0 | 2.78 | 0 | 0.70 ± 1.20 | Polluted |
| Water Quality Index, WQI | 47.44 | 36.32 | 44.78 | 47.57 | 44.03 ± 4.59 | Polluted |

The result shows that the average pH of both river water samples (upstream and downstream) can be categorized as Class I, where the water quality level is necessary to sustain the macro-aquatic life. However, the average pH downstream of the river was slightly higher than the river upstream possibly due to the discharge of the alkaline treated leachate.

The average dissolved oxygen content in the river water sample downstream was higher than that of upstream. The low DO content for both river samples (upstream and downstream) is due to the nitrification of ammonia [53], as the average ammonia nitrogen content for both can be categorized as Class V.

The average BOD$_5$ concentration in the river water samples upstream and downstream can be categorized as Class I and Class II, respectively. The average COD concentration at both upstream and downstream can be categorized as Class III, but the average COD content downstream is much higher than that of upstream. The COD value indicates the river downstream having higher organic matter content compared to the river upstream.

According to the sub-index result calculated from the equation in Table 3, the SIBOD values for both upstream and downstream can be categorized as clean, in the range of 91–100; but the SIAN values of both samples can be categorized as polluted, in the range of 0–70. The overall WQI both upstream and downstream can be categorized as polluted, indicating that Jinjang River is polluted.

### 3.5. Impact of Discharged Treated Leachate on Water Quality of Jinjang River

The parameter, such as in-situ pH, in-situ DO, BOD, COD, ammonia, and the TKN concentration upstream of the river, the treated leachate, and downstream of the river are shown in Figure 9a–f, respectively.

From Figure 9a, the results show the in-situ pH for all three samples being in the range of pH 6.5 to pH 8.65. However, the treated leachate samples, which are the final discharge from the leachate treatment plant, had the slightly highest pH except for the first sample, followed by the downstream samples and lastly, the upstream samples. The treated leachate, which had a high pH value, when discharged into the river, may lead to slightly higher pH downstream of the river than the upstream river.

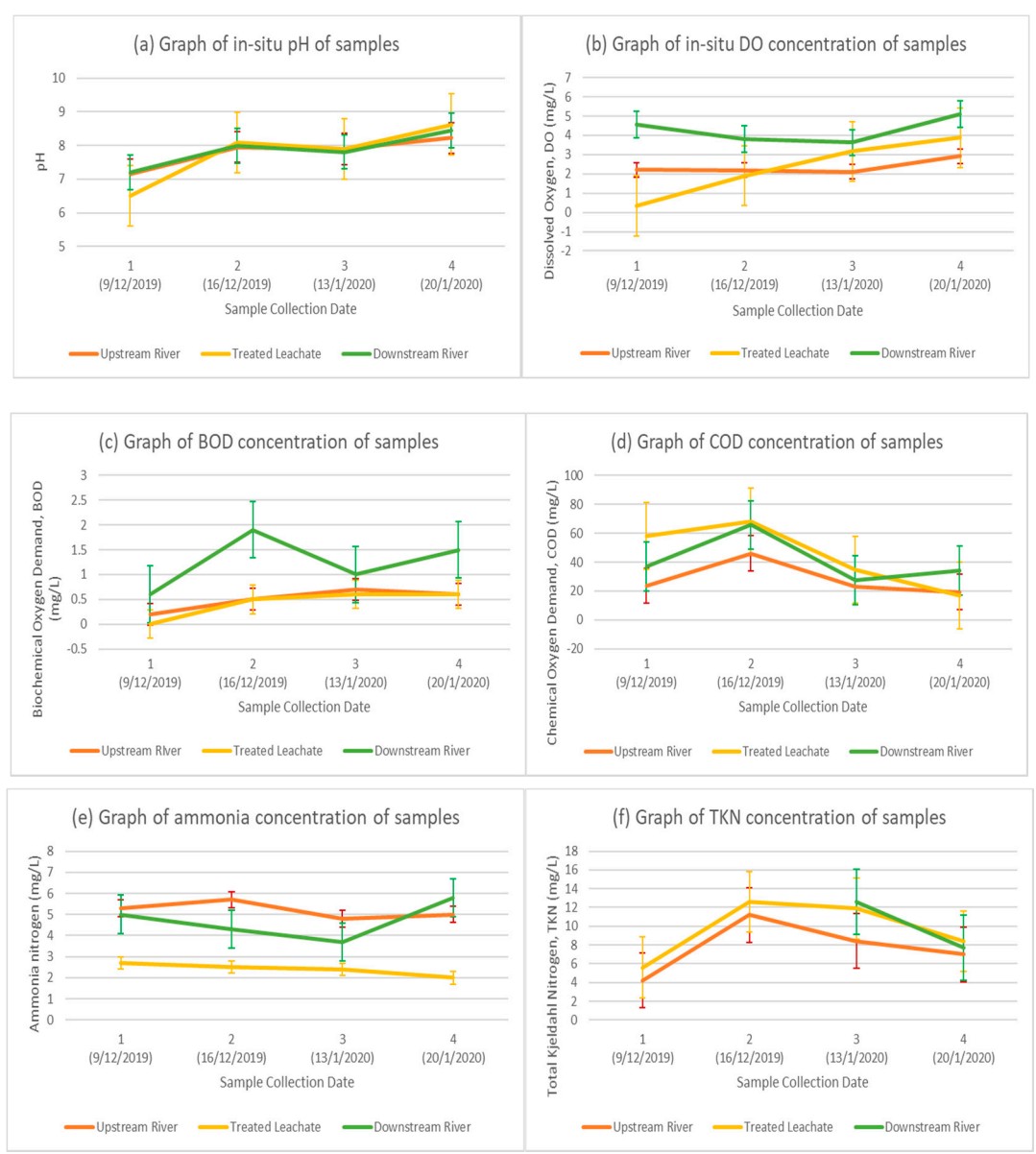

**Figure 9.** Graph of concentration of sample (**a**) in-situ pH, (**b**) in-situ DO, (**c**) BOD, (**d**) COD, (**e**) ammonia nitrogen, (**f**) TKN.

Figure 9b shows the downstream river having the highest concentration of dissolved oxygen. The lower DO concentration upstream may be due to the higher ammonia concentration, as shown in Figure 9e, due to the oxygen consumption via nitrification. Nitrogen can contribute to algae growth. As the algae die and decompose, the process consumes dissolved oxygen, thus resulting in insufficient amounts of dissolved oxygen available for aquatic life [54]. The source of nitrogen includes the discharge from wastewater treatment and greywater, as there is a Sewage Treatment Plant (STP) and a wet market upstream of the river. However, the treated leachate had the highest organic content because the LTP could not efficiently remove TKN.

The result shows that the BOD content downstream was the highest of the three samples, as shown in Figure 9c. The organic matter content in downstream may be slightly attributed to the treated leachate even if the BOD value is low. Besides, the ex-landfill in Taman Beringin located downstream may contribute to the discharge of raw leachate, even with the landfill being closed. The raw leachate from ex-landfill has high concentrations of organic matter as proven in a past study [8]. However,

the treated leachate had the highest concentration of COD and could have contributed to the increased COD downstream of the river compared to the river upstream, as shown in Figure 9d.

## 4. Conclusions

Based on the laboratory results and analysis, the raw leachate had high amounts of organic matter and toxic content. The raw leachate was categorized as intermediate leachate in the acetogenic and methanogenic phases. The BOD (697.50 ± 127.94 mg/L), COD (2419.75 ± 1155.22 mg/L), TSS (2710.00 ± 334.79 mg/L), and ammonia (317.08 ± 35.45 mg/L) of the raw leachate exceed the standard limits of the Environmental Quality (Control of Pollution from Solid Waste Transfer Station and Landfill) Regulation 2009 except for pH (7.73 ± 0.08) and temperature (32.7 ± 0.79 °C). Therefore, raw leachate must be treated completely before being discharged into the river to avoid the contamination.

The overall performance of the leachate treatment plant is satisfactory, as the final treated leachate discharged into the Jinjang River was able to meet the standard requirements of the Environmental Quality (Control of Pollution from Solid Waste Transfer Station and Landfill) Regulation 2009. The physicochemical process, such as coagulation, flocculation, and adsorption of activated carbon, has the biggest effect on whether the leachate can be treated effectively or not.

The performance of the LTP's activated sludge system was not satisfactory across several aspects such as in the removal of pollutants, operating parameters, design parameters, and kinetic parameters. The overall pollutant removal percentage did not reach 50%. The operational parameters, such as pH (8.4 ± 0.07), DO (0.16 ± 0.02 mg/L), and COD:N ratio (100:13) did not meet the standard requirement for maintaining the activated sludge system efficiency. The design parameters such as F/M ratio (0.13), SRT (2.69 days), and organic loading rate (0.68kg COD days/m$^3$) also did not reach the optimal rate. The kinetic parameter such as Y (5.655VSS, BOD$_5$ mg/L), k$_d$ (−0.247day$^{-1}$), $\mu_{max}$ (500 day$^{-1}$), and K$_s$ (−0.005 mg/L, BOD$_5$) affected the performance of activated sludge system. Overall, the result showed that the activated sludge system was not functioning well and the biological treatment alone was not sufficient to treat the leachate. The LTP's activated sludge in the Taman Beringin Solid Waste Transfer Station must be upgraded and maintained to reduce the cost of the chemical treatment.

The Jinjang River could be categorized as polluted based on the laboratory result. The discharge of the final treated leachate from LTP at Taman Beringin Solid Waste Transfer Station may contribute to river pollution. However, other factors also affect the river water quality, such as the sewage treatment plant upstream. The STP discharges wastewater into the river and may lead to increased biological pollutants, toxic chemical compounds, and other pollutants in the river. The discharged leachate from the ex-landfill located at downstream may mix with the surface waters and could also be one of the factors that pollute Jinjang River.

A few recommendations are suggested to improve the performance of the activated sludge system in treating the leachate. Based on this study, the pH value should be reduced to pH 7.5 and the DO content should be increased to 2.4 mg/L. The F/M ratio should be increased to 0.2–0.4; the SRT should be extended up to 20 days; and the OLR should be increased to 0.75–1.5 kg COD days/m$^3$ by increasing the influent flow or increasing the substrate. For the kinetic parameters, the Y value should be lowered and k$_d$ value should be increased to reduce the sludge production; the $\mu_{max}$ value should be lowered and the K$_s$ value should be higher to optimize the bacterial growth rate for substrate decomposition.

**Author Contributions:** P.M.C. and A.N.N. performed the experiment; P.M.C., A.N.N., and F.S. analyzed the data; M.F.A.U. provided the source of information for data analysis; all authors contributed to preparing the manuscript. All authors have read and agreed to the published version of the manuscript.

**Funding:** This research received funding from FRGS/1/2013/TK07/UKM/02/5 and LRGS MRUN/F2/01/2019.

**Conflicts of Interest:** The authors declare no conflict of interest.

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
