# Peer review of "Impact of Effluent from the Leachate Treatment Plant of Taman Beringin Solid Waste Transfer Station on the Quality of Jinjang River"

_processes, doi:10.3390/pr8121553_

Round 1
Reviewer 1 Report
The manuscript deals with a relevant topic in the field of environmental protection and monitoring. However, some significant issues hinder publication in the present form.
First, language is accurate only for two sections, namely “Abstract” and “Introduction”. A number of incorrect sentences as well as typos affects the rest of the manuscript.
In addition, a number of key aspects hinder publication at least at this stage.
The manuscript title refers to the evaluation of the potential impact of the effluent from the leachate treatment plant on the quality of the receiving water body. However, very little relevance is given to correctly develop this pivotal aspect. In literature, a number of models are available to study and predict the potential interaction between WWTP discharge and the receiving water body. On which basis, the correlation between effluent and receiving water properties was set, it’s not clearly explained.
In addition to this, manuscript title should be imporved, for example by crossing out from “from” to “plant” and replacing the text with: “Impact of the effluent from a leachate treatment plant on the quality of….”. Please, clearly state in the abstract that the leachate is a liquid residue produced by solid waste squeezing in a Mechanical treatment plant for municipal solid.
Reviewer’s impression is that Authors tried to develop (1) a generic analysis of the operating conditions of the plant, (2) a rough characterization of river, without going in depth for analysing and theoretically interpret the interaction between the two.
Please note that each of the two topic is broad and needs appropriate methodology to be corrected analysed. As an example, the analysis carried out on each stage of the WWTP is not fully justified, as the sampling campaign is not fully described. It is not clear, for example, whether each of the analysed sample is representative of a specific inlet to the leachate treatment plant.
The parameters analysed to evaluate the effluent quality are partial. In order to better understand the composition of the leachate and the evolution over the treatment train, particulate vs soluble COD is recommended, as well as metals and organics typically found in landfill leachate. Moreover, why did they analysed ammonia only (and in some cases TKN) neglecting nitrite and nitrates?
Moreover: is the leachate treatment plant operated in continuous? Which is the waterflow reaching the river? How can be evaluated the dilution factor, if any? Can the Authors justify the fact that river water was sampled just in two months? How such a sample strategy may take into account the different flowrates (i.e., the different vulnerability) during the year?
A number of assumptions are not clearly stated. For example, the low DO content of the leachate (line 147) can be explained by the metabolic reactions of aerobic microorganisms degrading the rapidly biodegradable matter in the leachate.
Treatment plant is described with not sufficient clarity. In the abstract, Authors stated that the biological treatment was followed by coagulation and flocculation, the latter two treatments allowing for an overall good performance of the whole process. By inspecting Figure 3, the treatment sequence appears to be different, as well as by reading the related text.
Lien 128 to 129: Authors must explain how data were fitted to derive such values. Please, note that info reported in Figure 7 and 8 and Table 8 are not sufficiently explanatory. Moreover, in Table 8 units of Y are wrong.
Equation 1, 2 and 3 are not properly visible in pdf. However, lbs cannot be used. Please, refer to SI.
Table 3 and related text: Please, explain the meaning of the term “sub-index” the first time You use it. Please, explain the formulas reported in the text, explain the meaning of “x”. For example, in the first raw, is “x” the DO? If so, please add the units, and briefly explain the reason of the numeric constraint (is 8 the maximum solubility value in water? At which temperature? Is distilled water? Is an in situ value?please, note that this is just an example. The whole table is hard to be read).
Figure 6: Please, add standard deviation for each measured point.
Please, explain why activated carbon may remove so high percentage of the whole set of contaminants considered by the Authors.
Line 190: pH values reported in the text do not match table 7 values.
Line 193 to 195: the explanation is vaguely reported and not so useful to demonstrate how the effluent impact the quality of the river. Moreover, it is not supported by data, as no information on the nitrification yields can be derived from the text.
The whole section 3 is not fairly detailed and clearly structured. Data appears to be not fully sufficient to derive the stated conclusions.
In general, the contribution to advance knowledge beyond the state of the art in the field of the manuscript is not sufficiently highlighted and justified.
Reviewer 2 Report
The paper is concerned with the impact of discharged leachate from Taman Beringin Solid Waste Transfer Station on the water quality of Jinjang River. The paper is well written and covers some important issues, so I recommend it for publication in Processes.
Detailed comments:
- The authors should underline the novelty of research in the Introduction.
- Have the authors done reproducible studies? If so, they should show the error bars in the figures.
- The discussion of obtained results with reference to be published so far paper is poor, so the manuscript needs to be improved very carefully.
- Figures 2 and 3 are of poor quality. Please insert Figures in high-resolution.
Reviewer 3 Report
The manuscript is presenting an interesting topic of water quality of Jinjang River and the performance of the LTP of Taman Beringin Solid Waste Transfer Station. Always the water quality issue nowadays is crucial, mostly due to increasing water scarcity. The topic of the manuscript is also in the scope of the journal.
The performance of activated sludge system of the leachate treatment plant at Taman Beringin Solid Waste Transfer Station was being evaluated through the percentages of removal of pollutants from leachate, operational parameters, design parameters and kinetic parameters. The physical and chemical properties of raw leachate were also being identified in this study.
Taking into account of the importance of the topic, the aim if the work should be better highlighted. Furthermore, the research gaps should be pointed out.
For the figures 1-5 the copyrights must be presented. In the situation when the authors took scheme/figure from other sources not only the reference but also the information about the copyright should appear in the figure capture.
Section 2.2 - Why only 1h difference was between the sample collection? Was it related to the special treatment or process in the unit itself?
How the samples were stored? Any preservatives were added?
Please use only the SI units. In the equations -1-3 there are pound instead of kg or for example.
Figure 6-9 must be improved. The quality of the presented figures is not suitable for publishing in the journal.
Figures 6 and 9 – do not connect the points. The current version suggests that reading o of the results were in continues mode. But only a few points were collected. Please correct and add error bars.
Generally, I would suggest to strength the discussion with the comparison with the works of others. The critical discussion should be presented.
The selected literature is suitable to the topic of the work and is really updated, ca.60% of the references from the last 5 years.
